# Quality of Life of Patients with Arterial Hypertension

**DOI:** 10.3390/medicina56090459

**Published:** 2020-09-09

**Authors:** Katarzyna Snarska, Monika Chorąży, Michał Szczepański, Marzena Wojewódzka-Żelezniakowicz, Jerzy Robert Ładny

**Affiliations:** 1Department of Clinical Medicine, Medical University of Bialystok, Jana Kilińskiego 1, 15-089 Bialystok, Poland; 2Department of Neurology, Medical University of Bialystok, Jana Kilińskiego 1, 15-089 Bialystok, Poland; chorazym@op.pl (M.C.); mics666@gmail.com (M.S.); 3Hospital Emergency Department with Intensive Care Unit and Cytostatics Delivery Point, Medical University of Bialystok, Jana Kilińskiego 1, 15-089 Bialystok, Poland; wojewodzkam@wp.pl; 4Department of General and Endocrine Surgery, Medical University of Bialystok, Jana Kilińskiego 1, 15-089 Bialystok, Poland; ladnyjr@wp.pl

**Keywords:** arterial hypertension, quality of life, WHOQOL-BREF scale, Barthel scale

## Abstract

Symptoms of hypertension with accompanying complications result in a significant reduction in patients’ quality of life. Effective conduct of prescribed pharmacotherapy supported by a healthy lifestyle allows to achieve satisfactory effects of treatment, which translates into an improvement in the quality of life of patients. The aim of the work was to determine the quality of life of patients with hypertension and the factors affecting it. The study included 100 people with hypertension, who are patients of the department of internal diseases of the hospital in Hajnówka during the period 1.6.2019–1.12.2019. The questionnaire survey, the standardized WHO Quality of Life (WHOQOL)-BREF scale and the Barthel scale were the research tools. The probability *p* < 0.05 was assumed as the level of statistical significance. The study group consisted of subjects between 30–89 years old. The majority were men and those living in the city. The average BMI (body mass index) of the subjects was 28.4 kg/m^2^. The duration of the disease among those surveyed was on average 7 ± 6.34 years. The highest-rated area of quality of life was the physical field and the lowest social sphere according to the WHOQOL-BREF questionnaire. Patients with hypertension have determined their quality of life at a good or medium level in the physical, psychological, social, and environmental sphere. There are many factors that improve quality of life in all areas. These include following the recommendations on modifiable risk factors.

## 1. Introduction

For two decades, the growing number of cases of arterial hypertension has been observed. Global data shows incidence at the level of 30–38% among adults, mostly in low and medium-developed countries. Numerous screening studies over the past decade have shown prevalence of arterial hypertension between 29–45% of the adult population, showing an increasing morbidity trend. Standardized prevalence rate in podlaskie province in 2018 was 296 cases per thousand inhabitants. It was the third lowest score in the country [1]. The highest morbidity is among the population over 65, both in Poland and worldwide. It affects 75% of seniors [2].

Symptoms are imperceptible or interpreted as the result of other conditions in the initial phase of arterial hypertension. It can cause pain and dizziness, nosebleeds, decreased exercise tolerance, increased tiredness, progressive shortness of breath, mood disorders, and impaired concentration. Symptoms specific to affected organ or system may appear as the untreated or incorrectly treated disease progresses [3].

Arterial hypertension is a major risk factor of cardiovascular diseases and kidney dysfunction. The increase in blood pressure (BP) correlates with the number of premature deaths. Hypertension with accompanying complications significantly decreases the quality of life of patients [1,4]. The aim of determining total cardiovascular risk is to make the best use of cardiovascular disease prevention measures according to the expert opinion of the European Society of Hypertension (ESH) and the European Society of Cardiology (ESC) [5]. Non-pharmacological treatment and pharmacotherapy implementation is particularly important in young patients with relevant risk factors. The main goal of treatment of arterial hypertension is achieving maximum reduction of the overall long-term risk of cardiovascular diseases. This action requires the treatment of arterial hypertension, as well as the correction of all concomitant modifiable risk factors, including smoking cessation, lowering total cholesterol level, and weight reduction through change in dietary habits and physical activity [4,5].

The aim of the work was to assess the quality of life and the factors affecting it in patients with arterial hypertension.

## 2. Material and Methods

### 2.1. Design and Participants

The study included 100 participants with arterial hypertension, who were hospitalized in the Department of Internal Diseases of Public Hospital in Hajnówka between 6 January 2019 and 12 January 2019. The research was conducted using medical records and a survey of our own authorship, the standardized WHO Quality of Life (WHOQOL)-BREF scale and the Barthel scale. Questionnaire was anonymous. Criteria for the inclusion of patients in the study: hypertension and informed consent to participate in the study. The exclusion criterion was lack of consent to participate in the study.
**The WHOQOL-BREF scale** consists of 26 questions, the first two of which serve a subjective assessment of the quality of life and state of one’s health. Another 24 questions, arranged in mixed order, serve to evaluate four domains of quality of life: physical, psychological, social, and environmental. The physical domain refers to the feeling of pain, the need for medications and treatment, the sense of satisfaction with daily performance in life and work. The psychological domain is used to assess the sense of complacency with life and external appearance. The social field describes relationships and support received. The environmental domain concerns the sense of security, housing and material conditions, the realization of interests, and communication [6].**The Barthel scale** is used to measure performance in activities of daily living (ADL). It contains 10 categories of questions where the ADL is scored at 0, 5, or 10 points. Ten variables describing ADL and mobility are scored, a higher number being a reflection of greater ability to function independently. Time taken and physical assistance required to perform each item are used in determining the assigned value of each item. The Barthel Index measures the degree of assistance required by an individual on 10 items of mobility and self-care ADL.The categories of self-reliance in the implementation of physiological needs, care, feeding, mobility and dressing are assessed. The scale score is between 0–100. The resulting sum of points allows you to define the level of incapacity in the following categories: benign (85–100 points)—indicates the patient’s self-reliance; medium-heavy condition (21–85 points)—indicates a dysfunction requiring partial assistance; very severe condition (≤20 points)—indicates a very large movement deficit, preventing the implementation of self-service activities [7,8].

### 2.2. Procedure and Ethical Considerations

The study was conducted between 6 January 2019 and 12 January 2019. The research conformed to Good Clinical Practice guidelines, and the followed procedures were in accordance with the Helsinki Declaration. All patients signed a consent form to participate in the study. The research was approved on 28.12.2018 by the Bioethics Committee of the Medical College in Bialystok (consent number: KB/63/2018/2019).

### 2.3. Statistical Analysis

The studies were anonymous and voluntary. They were subjected to descriptive, graphic and statistical analysis. The calculation was done using Microsoft Excel 2016 and the STATISTICA v21.0 SPSS (New York, NY, USA) statistical package. The results obtained were statistically analyzed by χ^2^ test for independent samples. A 5% risk of inference error was assumed. A chi square test was used to examine the statistical relationship between the analyzed characteristics. The probability *p* < 0.05 was assumed as the level of statistical significance.

## 3. Results

### 3.1. Characteristics of the Study Population

The study group were 30–89 years old, with an average age of 63. Mean BMI (body mass index) in the group was 28.4 kg/m^2^. Men and people living in the city were the majority (57%). Housing conditions were assessed by respondents as good and very good. Most often, the care of the respondent was carried out by a spouse. The source of livelihood was mainly retirement pension. The highest percentage of respondents was of secondary education (Table 1).

The duration of the disease among those surveyed ranged from 3 months to 28 years (on average 7± 6.34 years). The mean systolic pressure values were 140.45 ± 10.79 mmHg, and diastolic pressure values were 89 ± 9.48 mmHg. Not all respondents used visits to the outpatient hypertension treatment clinic. Treatment was most often based on taking two or one hypotensive drug. Among respondents, 87% had their own blood pressure measuring apparatus, and the average number of measurements was less than 2 per day (Table 2).

Among concomitant diseases, respondents listed ischemic heart disease (27%), myocardial insufficiency (16%), osteoarthritis (14%), type 2 diabetes (9%), atrial fibrillation (7%), chronic kidney disease (3%), and atherosclerosis (1%).

Among the predisposing factors for hypertension, respondents were most likely to list over-salting of meals (75%), coffee consumption (71%), alcohol consumption (45%), smoking (41%), consumption of canned products (41%), lack of physical activity (34%), low water intake (11%), sweetening of drinks (4%), and consumption of sweet soft drinks (2%).

The vast majority—75%—of patients were slightly disabled according to the Barthel scale, while 24% were moderately disabled. None of the respondents were classified as very severe disability.

### 3.2. Quality of Life Assessment

The highest-rated domain of quality of life, according to the WHOQOL-BREF questionnaire, was the physical health domain, the lowest—quality of life in the social domain. The statistical analysis did not show a significant relationship in the quality of life assessment in each area (Table 3).

Among both sexes, the quality of life was rated at most in the physical domain and the lowest in the social domain. The statistical analysis did not show a significant relationship between the gender of the respondents and the quality of life assessment in each domain (Table 4).

The physical domain of quality of life was assessed at the highest, and the social domain was the lowest—both among the inhabitants of towns and villages (NS).

Patients assessing housing as very good were those that rated their quality of life in the psychological domain the highest. Patients assessing housing as good rated the best quality of life in the physical domain and lowest in the social domain (NS).

The respondents, who were cared for by a spouse or children, rated the physical domain at the highest level and the social domain at the lowest. Those who were cared for by their son-in-law or daughter-in-law rated the quality of life in the physical and social domain at the highest and the environmental domain at the lowest (NS). Those living independently gave highest notes in the social and environmental domain and the lowest in the psychological (NS).

When taking livelihoods into account, all patients gave highest notes in the physical domain and the lowest in the social domain. The range of scores in individual domains among those who maintained their work ranged between 12.82 and 13.38. The statistical analysis showed a significant relationship between the source of income and the quality of life assessment in all domains—physical (*p* = 0.005), psychological (*p* = 0.003), social (*p* = 0.033), and environmental (*p* = 0.0001).

The highest-rated domain of quality of life was the physical, and the lowest was social domain among those with primary, secondary, and vocational education. Respondents with university education rated the social sphere of quality of life at the highest level and the lowest psychological sphere. Average scoring ranges among people with primary education ranged between 15.48 and 16.35, while, among patients with tertiary education, they were between 12.12 and 12.62. Average scoring ranges among people with primary education ranged between 15.48 and 16.35, while, among patients with tertiary education, they were between 12.12 and 12.62. The statistical analysis showed a significant relationship between education and quality of life assessment in the physical (*p* = 0.001), psychological (*p* = 0.001), social (*p* = 0.001), and environmental domain (*p* = 0.001) (Table 5).

The physical domain of quality of life was highest rated by patients treated with 1 or 2 hypotensive medicines and lowest rated by those not taking hypotensive medicines. The social domain was highest rated by patients not taking hypotensive drugs and the lowest by those taking 1 or 2 hypotensive drugs. Patients treated with 3 hypotensive drugs rated the physical and environmental domains at the highest level and psychological one the lowest. The statistical analysis did not show a significant relationship between the hypotensive drugs intake of the respondents and the quality of life assessment in each domain. The highest-rated field of quality of life was the physical sphere, the lowest—social among patients owning sphygmomanometer. Average scores ranged from 13.93 to 14.9. The statistical analysis showed a significant relationship between possessing sphygmomanometer by the respondents and quality of life assessment in the physical (*p* = 0.017), psychological (*p* = 0.01), social (*p* = 0.009), and environmental domain (*p* = 0.002) (Table 6).

The physical domain of quality of life was highest rated by all patients with concomitant chronic diseases. Respondents with continuous atrial fibrillation rated the psychological domain equally highly, while atherosclerosis sufferers gave highest notes to the psychological and environmental domain. The social domain was worst rated unanimously. The statistical analysis did not show a significant relationship between the occurrence of concomitant diseases in the respondents and the quality of life assessment in each domain (Table 7).

Patients who consume alcohol, canned food, food sweeteners, and do not engage in physical activity and reduce water consumption rated the poorest quality of life in the social domain. Patients consuming coffee rated the environmental domain of quality of life the lowest. The average score was highest in the physical domain, regardless of the type of risk factors. The statistical analysis showed a significant relationship between existing risk factors for hypertension and quality of life in the psychological (*p* = 0.021), social (*p* = 0.03), and environmental domain (*p* = 0.001) (Table 8).

Patients who considered stress to be the cause of the increase in blood pressure gave highest notes to the physical domain of quality of life area. People who indicated physical exertion or fatigue as the cause of hypertension gave highest notes to the psychological domain. The social domain achieved the lowest average number of points among all respondents, regardless of what reason they indicated as the cause of hypertension. The statistical analysis showed a significant relationship between the causes of the increase in blood pressure in the respondents and the quality of life assessment in the social domain (*p* = 0.002) (Table 9).

In terms of symptoms of hypertension, all respondents awarded the highest number of points in the physical domain of quality of life and the least in the social domain (NS). (Table 10).

The highest-rated domain of quality of life was the physical domain, the lowest – social among those surveyed with a slight disability, according to the Barthel scale, and the range of average scores in each domain was 14.53–15.49. Patients with medium-severe disability have given the most points in the environmental domain, the least in the social, and the range of average scores in each field was between 10.96 and 12.67. The statistical analysis showed a significant relationship between the level of disability of the respondents and the assessment of quality of life in the physical (*p* = 0001), psychological (*p* = 001), social (*p* = 0.0001), and environmental domain (*p* = 0.0011) (Table 11).

## 4. Discussion

Arterial hypertension is a chronic disease. It poses a serious problem for contemporary medicine despite the constant improvement of therapeutic methods. It is the main cause of cardiovascular and kidney diseases and significantly affects mortality. The presence of hypertension is associated with lower Health-Related Quality of Life [2,9,10,11,12]. The aim of modern medicine is not only to prolong life of a patient with chronic disease but also to bring his quality of life as close as possible to the state before the disease. Studies on quality of life show that the quality of life experienced by patients depends on the causes, symptoms, and therapies of existing diseases. The WHO emphasizes that a sense of quality of life is not limited to general well-being or the lack of somatic ailments. The WHOQOL-BREF scale introduced by WHO enables thorough analysis of the patient’s quality of life and individual state of health, especially in specific areas contributing to a holistic picture of quality of life: physical, mental, social, and environmental [13,14,15].

The studies aiming to assess the quality of life in hypertensive patients were conducted by many authors from all over the world—China, India, Brazil, USA, and Poland. The results unequivocally indicate that hypertension occurrence significantly influences the decrease in a patient’s quality of life [14,16,17,18].

Considering different areas of quality of life, respondents of our study gave highest scores in physical dimension of life, and the lowest score in social. The obtained results differ from data from other authors, who showed decrease in the physical dimension of life [13,14,15,19]. The participants of study conducted in China showed significant decrease in the quality of life both in physical, as well as social, areas [11]. Depending on the research group selection, the highest scores of quality of life were obtained in social [14], as well as psychological, areas [13,20].

Many authors point at the specific factors that contribute to the decrease in quality of life in patients with hypertension. Foreign reports show a significant impact of age, female sex, duration of the disease, number of medications taken, and systolic pressure as predictive factors in the quality of life of patients with hypertension [16,17,21]. Polish studies, conducted by Sawicka research group, also showed a significant correlation of deterioration in quality of life in people over 60 years of age and male sex [14]; in other national studies, women showed a lower sense of life quality [18,19]. Better quality of life was shown among men in Vietnam, while the decrease in quality of life was dependent on higher education and older age, and rural residence corresponded only with decrease in the psychological domain [8]. Greek research also shows that a poorer sense of quality of life is closely linked to the female sex and older age [21]. Improvement in quality of life can be expected in patients with short-term hypertension, mainly in young people according to Kawecka-Jaszcz. Both the duration of the disease and age correlated with a worse quality of life of the subjects [22]. Our study showed no significant differences in the assessment of each area of quality of life based on gender, place of residence, housing, people caring for the respondent, the amount of hypotensive drugs taken, concomitant chronic diseases, and symptoms of hypertension. These results are different from those obtained by other authors, who showed a decrease in quality of life in the physical and mental domain along with the number of medicines taken, which is probably due to polytherapy side effects [14,23,24,25]. Adherence to pharmacological treatment has a positive impact on the mental and physical domains of patients, as it did on the overall quality of life score in meta-analysis conducted by Portuguese authors [26].

The source of income had a significant impact on the assessment of all areas of quality of life of those surveyed. Hypertension symptoms and the need of regular pharmacotherapy significantly impair functioning of those who work. Those living from pension at home have more capability to self-control, which undoubtedly has a beneficial effect on the physical functioning of the body, mental well-being and social and environmental relationships [27]. Different results in physical domain were obtained in the Sawicka study, where the best quality of life was presented by working people and the worst by pensioners. Other domains did not demonstrate significant dependencies [14].

Education was an important factor in assessing all domains of quality of life. People with lower education level may have lower expectations than well-educated patients, which affects the level of perceived quality of life in all domains. These results differ from the results of Asian studies, where higher education and good economic condition significantly increased the sense of quality of life [8,21].

Owning an BP measurement device has a positive effect on a sense of quality of life in all domains. Owning a sphygmomanometer gives a sense of self-control, but it also allows to react in time to increased blood pressure and thus improve well-being [23].

Practicing behavior conducive to hypertension has significantly influenced the assessment of quality of life in the psychological, social and environmental domains. Of the factors mentioned, the highest quality of life scores were given by respondents who sweetened drinks and consumed sweet soft drinks. When consuming sweetened drinks, patients may in a sense feel that they “sweeten” their lives, thus improving its quality. The consumption of canned products correlated with the lowest quality of life assessment values. Salt-rich food has a direct effect on increasing endovascular pressure, which affects the course of hypertension, consequently resulting in decreased sense of quality of life [24,28].

Situations that cause increase in blood pressure have a significant impact on the assessment of the social domain of quality of life [25]. The highest notes on social domain of quality of life were given by the subjects who indicated stress as the cause. They are probably more aware of the real dangers, so they can learn to deal with them or avoid them. Subjects who perceive physical exertion or fatigue as the causes of the increased blood pressure do not see actual risks, unknowingly exposing themselves to risk of hypertension [29]. Avoiding physical exercise and fatigue contributes to the reduction of social contacts, which translates into a decrease in the social domain of quality of life.

## 5. Limitation Section

Our study is somewhat limited by the quantity of population for statistical measurement. Some limitations can also arise from habits and cultural biases of local population.

## 6. Conclusions


The quality of life of patients with hypertension was assessed at a good, medium level in the physical, psychological, social, and environmental domain in the studied group.Significant positive effects on quality of life in all areas were: livelihood, education, and owning a blood pressure measurement device.The occurrence of factors conducive to hypertension, in particular the consumption of sweet soft drinks and sweeteners, increased the quality of life in the psychological, social, and environmental domains.Stress, fatigue, and physical exertion increased the assessment of the quality of life in the social domain.


## Figures and Tables

**Table 1 medicina-56-00459-t001:** Socio-demographic characteristics of the study population.

	Range/N	Mean ± SD/%
Age (years)	30–89	62.8 ± 14.95
BMI (kg/m^2^)	14.1–52.7	28.4 ± 6.36
Gender–female	43 (43%)	
Place of residence–city	57 (57%)	
Housing conditions		
Very good	3 (3%)	
Good	97 (97%)	
Bad	-	
Person taking care for the examinee		
Wife/husband	74 (74%)	
Daughter/son	21 (21%)	
Son-in-law/daughter-in-law	2 (2%)	
Other–alone	3 (3%)	
Livelihood		
Work	34 (34%)	
Retirement pension	61 (61%)	
Disablement pension	5 (5%)	
Other	-	
Education		
Basic	23 (23%)	
Secondary	46 (46%)	
Vocational	23 (23%)	
Higher	8 (8%)	

Abbreviations: BMI, body mass index; SD, standard deviation.

**Table 2 medicina-56-00459-t002:** Clinical characteristics of hypertension.

	Range/N	Mean ± SD/%
Duration of disease (years)	3 months–28 years	7.24 ± 6.34 years
BP values (mm/Hg)		
systolic	110–160 mmHg	140.45 ± 0.79 mmHg
diastolic	70–105 mmHg	89 ± 9.48 mmHg
Annual number of visits to outpatient hypertension treatment clinic	0–2	0.81 ± 0.59
Constantly taken hypotensive drugs		
0 medications	2 (2%)	
1 med	33 (33%)	
2 meds	52 (52%)	
3 meds	12 (12%)	
4 meds	1 (1%)	
Being in possession of pressure measuring apparatus	87 (87%)	
Daily quantity of self-contained BP measurements	0–3	1.71 ± 0.84

Abbreviations: BP, blood pressure.

**Table 3 medicina-56-00459-t003:** WHO Quality of Life (WHOQOL)-BREF assessment of the different domains of quality of life of the whole study group.

Quality of Life Domains				*p*
Physical Health	Psychological	Social Relationships	Environment	
14.69 ± 2.23	14.2 ± 2.26	13.67 ± 2.61	14.31 ± 1.73	0.221

**Table 4 medicina-56-00459-t004:** The impact of gender on the assessment of domains of quality of life.

	Quality of Life Domains			
	Physical Health	Psychological	Social Relationships	Environment
Female	14.58 ± 2.1	14.28 ± 2.21	13.46 ± 2.48	14.35 ± 1.64
Male	14.77 ± 2.32	14.14 ± 2.29	13.82 ± 2.69	14.28 ± 1.79
*p*	0.618	0.892	0.414	0.525

**Table 5 medicina-56-00459-t005:** The impact of the general characteristic factors on the assessment of the different areas of quality of life.

			Quality of Life Domains			
			Physical	Psychological	Social	Environmental
Place of living		City	14.3 ± 2.23	13.89 ± 2.2	13.26 ± 2.77	14.14 ± 1.65
		Country	15.21 ± 2.13	14.6 ± 2.27	14.21 ± 2.26	14.53 ± 1.81
***p***			**0.044**	**0.122**	**0.074**	**0.263**
Housing conditions	Very good		14.33 ± 1.88	15.33 ± 0.47	14.33 ± 2.36	14.33 ± 1.25
	Good		14.7 ± 2.24	14.16 ± 2.28	13.65 ± 2.61	14.31 ± 1.74
***p***			**0.782**	**0.383**	**0.659**	**0.981**
Care provider	Wife/Husband		14.4 ± 2.32	13.94 ± 2.33	13.34 ± 2.69	14.09 ± 1.78
	Daughter/Son		15.24 ± 1.69	14.95 ± 1.84	14.33 ± 1.96	14.81 ± 1.47
	Son-in-law/Daughter-in-law		17.5 ± 0.5	17	17.5 ± 1.5	16.5 ± 0.5
	Self-sufficient		16 ± 1.41	13.33 ± 0.47	14.67 ± 188	14.67 ± 0.47
***p***			**0.084**	**0.079**	**0.062**	**0.102**
Livelihood	Work		13.7 ± 2.47	13.18 ± 2.42	12.82 ± 2.87	13.38 ± 1.86
	Retirement pension		15.24 ± 1.9	14.8 ± 2.01	14.21 ± 2.29	14.87 ± 1.44
	Disablement pension		14.6 ± 1.96	13.8 ± 1.67	12.8 ± 2.56	13.8 ± 0.98
***p***			**0.005**	**0.003**	**0.033**	**0.0001**
Education	Primary		16.35 ± 1.49	15.83 ± 1.68	15.48 ± 1.56	15.52 ± 1.47
	Secondary		14.43 ± 2.22	13.98 ± 2.3	12.93 ± 2.73	14.06 ± 1.61
	Vocational		14.3 ± 1.92	13.74 ± 1.7	13.69 ± 2.46	14.22 ± 1.38
	Tertiary		12.5 ± 1.87	12.12 ± 2.02	12.62 ± 2.06	12.5 ± 1.73
***p***			**0.001**	**0.001**	**0.001**	**0.001**

**Table 6 medicina-56-00459-t006:** The impact of hypertension therapy factors on the assessment of specific domains of quality of life.

		Quality of Life Domains			
		Physical	Psychological	Social	Environmental
Constantly taking hypotensive drugs	0 meds	13	13 ± 1	15	13.5 ± 0.5
	1 med	14.09 ± 2.18	13.58 ± 2.06	12.61 ± 2.45	13.67 ± 1.73
	2 meds	15.11 ± 2.26	14.44 ± 2.44	14.04 ± 2.73	14.65 ± 1.7
	3 meds	14.83 ± 2.03	14.08 ± 1.63	14.67 ± 1.65	14.83 ± 1.4
	4 meds	14	15	15	13
***p***		**0.224**	**0.141**	**0.697**	**0.059**
Owning a BP measuring device	Yes	14.9 ± 2.14	14.42 ± 2.22	13.93 ± 2.54	14.52 ± 1.63
	No	13.31 ± 2.33	12.69 ± 1.68	11.92 ± 2.33	12.92 ± 1.73
***p***		**0.017**	**0.01**	**0.009**	**0.002**

**Table 7 medicina-56-00459-t007:** The impact of concomitant chronic diseases on the assessment of different domains of quality of life.

	Quality of Life Domains			
	Physical	Psychological	Social	Environmental
No other chronic diseases	13.3 ± 2.02	12.84 ± 2.16	12.13 ± 2.61	13.13 ± 1.42
Ischemic heart disease	14.85 ± 1.82	14.33 ± 1.65	14.07 ± 1.9	14.55 ± 1.2
Continuous atrial fibrillation	15.28 ± 2.43	15.28 ± 2.31	14.14 ± 2.69	14.71 ± 1.83
Atherosclerosis	13	13	9	13
Myocardial insufficiency	16.25 ± 0.8	16.12 ± 0.78	15.65 ± 1.9	15.87 ± 0.78
Type 2 diabetes.	14.89 ± 2.08	14.33 ± 1.05	14 ± 0.95	14.55 ± 0.95
Osteoarthritis	16.07 ± 1.98	15.21 ± 1.85	14.78 ± 2.01	15.21 ± 1.65
Chronic kidney disease	16 ± 1.41	15.67 ± 0.47	14.33 ± 0.94	15.33 ± 0.47
***p***	**0.697**	**0.222**	**0.197**	**0.921**

**Table 8 medicina-56-00459-t008:** The impact of factors conducive to hypertension on the assessment of different domains of quality of life.

	Quality of Life Domains			
	Physical	Psychological	Social	Environmental
Smoking	14.51 ± 2.54	13.58 ± 2.41	13.68 ± 2.89	14.17 ± 1.96
Coffee consumption	14.35 ± 2.34	13.96 ± 2.31	13.13 ± 2.73	14.03 ± 1.65
Alcohol consumption	14.67 ± 2.33	14.07 ± 2.45	14 ± 2.64	14.13 ± 1.93
Salting of dishes	14.61 ± 2.31	14.11 ± 2.37	13.57 ± 2.58	14.24 ± 1.72
Sweetening of drinks	15.75 ± 1.64	15 ± 1.22	14.25 ± 3.03	15 ± 1.22
Lack of physical activity	15.67 ± 1.79	15.5 ± 1.56	14.94 ± 1.97	15.23 ± 1.21
Consumption of sweet soft drinks	17	14.5 ± 1.5	16	15.5 ± 0.5
Consumption of canned products	13.9 ± 2.33	13.68 ± 2.45	12.9 ± 2.78	13.83 ± 1.71
Reducing water consumption	15.72	15.27 ± 1.48	13.72 ± 2.34	15.09 ± 1.68
***p***	**0.509**	**0.021**	**0.03**	**0.001**

**Table 9 medicina-56-00459-t009:** The impact of situations causing an increase in blood pressure on the assessment of specific domains of quality of life.

	Quality of Life			
	Physical	Psychological	Social	Environmental
Stress	13.76 ± 2.41	13.19 ± 2.31	12.78 ± 2.75	13.52 ± 1.84
Exercise	12.5 ± 0.5	13.5 ± 1.5	10 ± 1	13
Fatigue	12.5 ± 0.5	14 ± 1	11.5 ± 0.5	13
***p***	**0.277**	**0.119**	**0.002**	**0.551**

**Table 10 medicina-56-00459-t010:** Effects of the symptoms of hypertension on the assessment of the different domains of quality of life.

	Quality of Life Domains			
	Physical	Psychological	Social	Environmental
Tinnitus, headaches and dizziness, nosebleeds	14.71 ± 2.27	14.36 ± 2.21	13.89 ± 2.6	14.4 ± 1.58
Insomnia, redness of the face, cold sweats	14.1 ± 2.3	13.2 ± 2.82	12.9 ± 2.16	14 ± 2.68
Drowsiness, scotoma, chills	15.14 ± 2.29	14.28 ± 2.37	13.43 ± 3.11	14.28 ± 2.05
Do not know	14.87 ± 1.54	13.87 ± 1.16	12.75 ± 2.22	13.87 ± 0.93
***p***	**0.308**	**0.557**	**0.447**	**0.13**

**Table 11 medicina-56-00459-t011:** The impact of the degree of disability according to the Barthel scale on the assessment of the different domains of quality of life.

	Quality of Life Domains			
	Physical	Psychological	Social	Environmental
Slight disability	15.49 ± 1.79	14.87 ± 1.9	14.53 ± 2.1	14.83 ± 1.5
Medium-heavy grade of disability	12.17 ± 1.52	12.08 ± 1.98	10.96 ± 2.15	12.67 ± 1.34
***p***	**0.0001**	**0.001**	**0.0001**	**0.0011**

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
