# Peer review of "Quality of Life of Patients with Arterial Hypertension"

_medicina, 2020, doi:10.3390/medicina56090459_

Round 1

Reviewer 1 Report

Thank you for the opportunity to review your work again. 

The study included 100 people with hypertension, who were hospitalized in the Department between 01.06.2019 and 01.12.2019.  Was this all eligible patients during this time?

In reviewing your instruments, WHOQOL-BREF still raises questions. Add your permission to use the instrument and how to use the instrument.

In your used instruments, I still miss the validity and reliability of each instrument.

In the tables write p-values instead of P.

In table 8 and in line 317 change Coca-Cola to sweet soft drinks.

The paper is required to read by a native English speaker. For example line 256: You wrote “the research group selection”. The research group is the authors, I think the study population is better.

Add a limitation section in your discussion.

Please use Vancouver referencing style. The reference list must include the full information for all the works cited in the running text. The entries in the reference list should be placed in the same order in which they are cited in the text. Check also that all references in the list are in the manuscript. For example, I can not see nr 9.

Add same more keywords,

Author Response

Thank you for your favorable comments on our article.

The study included 100 people with hypertension, who were hospitalized in the Department between 01.06.2019 and 01.12.2019.  Was this all eligible patients during this time?

All the patients who gave their consent were included in the study. These were not all the patients hospitalized in the Department.

In reviewing your instruments, WHOQOL-BREF still raises questions. Add your permission to use the instrument and how to use the instrument. In your used instruments, I still miss the validity and reliability of each instrument.

The research was approved by the bioethical commission at the Medical University in Białystok. Consent number: KB / 63/2018/2019.

In the tables write p-values instead of P.

In the tables, p-values was entered instead of P.

 In table 8 and in line 317 change Coca-Cola to sweet soft drinks.

 Replaced Coca-Cola in table 8 and line 317 with sweet soft drinks.

The paper is required to read by a native English speaker. For example line 256: You wrote “the research group selection”. The research group is the authors, I think the study population is better.

Thank you for your attention. Is due to a typing error. In line 256: "study group selection" has been replaced with the study population.

Add a limitation section in your discussion.

Added a limitation section in the discussion. Our study is somewhat limited by the quantity of population for statistical measurement. Some limitations can also arise from habits and cultural biases of local population.

Please use Vancouver referencing style. The reference list must include the full information for all the works cited in the running text. The entries in the reference list should be placed in the same order in which they are cited in the text. Check also that all references in the list are in the manuscript. For example, I can not see nr 9.

Passwords were made. Item 9 and 10 included for download (editorial error).

Add same more keywords.

Keywords: arterial hypertension; quality of life, WHOQOL-BREF scale, Barthel scale.

Reviewer 2 Report

The authors have put in a lot of work on this manuscript and it is quite clear to read. Some comments to consider:

  1. The major issue is the stats analysis. It's recommended practice to use both multivariate & multivariable techniques, rather than a long list of univariate analyses. For example, the entry of multiple predictors would be more efficient and simplify the results greatly. Please consult with a stats expert.
  2. Please confirm the stats methods used- it doesn't appear to me that the chi-square was the only relevant test. 
  3. Given the sheer number of multiple comparison, some sort of correction for type 1 error would be needed. Again, the use of multvar. techniques would help to address this.  
  4. When discussing results, it would be best to focus only on interpreting sig. findings. In some cases, the overall test of sig. doesn't necessarily indicate which specific groups differ. 

Author Response

Thank you for your favorable comments on our article.

The major issue is the stats analysis. It's recommended practice to use both multivariate & multivariable techniques, rather than a long list of univariate analyses. For example, the entry of multiple predictors would be more efficient and simplify the results greatly. Please consult with a stats expert.

Please confirm the stats methods used- it doesn't appear to me that the chi-square was the only relevant test.

Given the sheer number of multiple comparison, some sort of correction for type 1 error would be needed. Again, the use of multvar. techniques would help to address this. 

When discussing results, it would be best to focus only on interpreting sig. findings. In some cases, the overall test of sig. doesn't necessarily indicate which specific groups differ.

The statistics used allowed to obtain answers to all research questions. However, it became unnecessary to extend them with more advanced statistical methods at this stage of the research. Nevertheless, we will consider this consideration in future research.

Round 2

Reviewer 2 Report

I understand that the research is at an early stage, and believe in the importance of publishing this work because it will bring attention to qol issues in hypertension patients in the authors' context. However the authors' response doesn't address my concerns. I am not asking for the application of overly complicated stats techniques that don't fit this situation. I am asking for the use of regression or anova when appropriate. The authors have also not addressed the specific question I had asked about the use of chi-square.

Author Response

After consulting with the statistician who performed the statistical analysis, we reply that in this study we used the ANOVA test due to the fact that we worked only on mediums.

This manuscript is a resubmission of an earlier submission. The following is a list of the peer review reports and author responses from that submission.

Round 1

Reviewer 1 Report

Thank you for the opportunity to review your work. 

10 of 24 references are in Polish and I know that a lot of matching references in English. Please provide more current references for your work.

Data collection, did you have a demographic, clinical date form or you used the records? In the Method section, you must add how you get the characteristics of the studied group.

The study included 100 people with hypertension, who were hospitalized in the Department between 01.06.2019 and 01.12.2019. Was it all eligible patients during this time? Describe the inclusion and exclusion criteria in the paper.

In reviewing your instruments,WHOQOL-BREF raises questions. Add a reference to the original work, your permission to use the instrument and how to use the instrument.

The Barthel scale also raises the question.  Add a reference to the original work. In the original work, you both have self-report but you also have direct observations and I can not see this in your method section. Neither I can see any description about who to score.

In your used instruments, validity, and reliability for each instrument should be stated.

.In results: I suggest a clarification regarding which instrument the results are based on.

The paper is required editing by a native English speaker. For example line 142: You wrote “Socio-demographic characteristics of the research group”. The research group is the authors, I think the study population is better. I think the word carer is wrong or really need all patients a carer? Maybe they need a support person?

Add a limitation section in your discussion section.